# OpenReview forum: "When Actions Go Off-Task: Detecting and Correcting Misaligned Actions in Computer-Use Agents"
_ICML.cc/2026/Conference — ICML 2026 regular_

### Official Review · Reviewer_LAzs · 2026-02-19

**Soundness:** 3
**Presentation:** 3
**Significance:** 4
**Originality:** 3
**Overall Recommendation:** 5
**Confidence:** 4

**Summary:**

This paper tackles a real problem: computer agents often drift off-track, either because they get tricked by attacks or just make bad decisions . Instead of just checking for safety violations, the authors focus on action misalignment. The work makes three primary contributions: (1) MISACTBENCH, a comprehensive benchmark with over 2,000 human-annotated action steps derived from both adversarial attacks and synthesized internal errors; (2) DEACTION, a runtime guardrail that employs a two-stage first Fast Check then Systematic Analysis pipeline. The extensive experiments showing that DEACTION outperforms existing baselines by over 15% in F1 score offline and reduces attack success rates by over 90% in online settings .

**Compliance With Llm Reviewing Policy:**

Affirmed.

**Final Justification:**

My final recommendation is to maintain my positive score of 5 for this paper. The authors' rebuttal successfully and fully addressed my main concerns.

Specifically, the detailed explanation of how the narrative summarizer implicitly mitigates cumulative drift was very clear and compelling, significantly strengthening the evaluation of the paper's clarity and soundness.


Although the authors' responses are exemplary, after carefully considering a potential score adjustment, I have concluded that a score of 5 remains the most accurate reflection of the paper's solid baseline contribution. The rebuttal has reinforced my prior positive assessment.

**Key Questions For Authors:**

1. Since latency is a critical bottleneck for real-time interactive agents, have you considered or experimented with fine-tuning a smaller, specialized open-source model for this alignment analysis to achieve both low latency and high accuracy?
2. Does the narrative summarizer provide any implicit defense against this long-term drift, or would detecting it require a shift to trajectory-level evaluation?

**Limitations:**

The authors adequately discuss technical limitations through a detailed error analysis (Section 5.1 and Appendix E.4) . Please consider adding a brief discussion on potential negative societal impacts or dual-use risks

**Strengths And Weaknesses:**

Strengths:
1. The paper makes a compelling argument for shifting the focus from safety constraints to broader action alignment. Categorizing both internal errors and external attacks captures real-world agent failures far better than prior frameworks.
2. MISACTBENCH fills a major gap in action-level evaluation . By combining real adversarial attacks with synthesized internal errors, it provides a highly reliable and much-needed dataset.
3. DEACTION effectively balances accuracy and latency . The two-stage pipeline and narrative history summaries keep it fast, while the iterative feedback actively fixes errors rather than just blocking them.

Weakness:
1. The system excels at detecting misalignments but struggles to accurately classify their exact type.
2. In Section 4, the authors mention that the guardrail evaluates actions exclusively at the step level. A key drawback of this isolated approach is the lack of global trajectory awareness, a series of seemingly harmless actions could bypass detection and cumulatively cause an off-task state.

---

> ### Author Rebuttal · Authors · 2026-03-31
>
> We sincerely thank the reviewer for the positive assessment and for recognizing that our focus on action misalignment **“captures real-world agent failures far better than prior frameworks”**, that MisActBench **“fills a major gap in action-level evaluation”** as a **“highly reliable and much-needed dataset”**, and that DeAction **“effectively balances accuracy and latency”** through its two-stage pipeline and iterative error correction.
>
> ### 1. Classification Accuracy (W1)
> We agree that fine-grained classification of misalignment types is more challenging than binary detection, particularly in our pre-execution setting: the guardrail must classify an action before execution, without access to the true outcome.
>
> As a result, distinguishing two internally arising categories requires accurately reasoning about expected consequences. This is considerably harder than identifying Malicious Instruction Following, which often relies on explicit external injections. In fact, this fine-grained per-category analysis is exactly what MisActBench makes measurable for the first time.
>
> However, for practical purpose of runtime guardrail, the critical decision is binary: whether to allow or intercept an action before execution. The three-category classification serves primarily as an analysis aid. Improving fine-grained classification for misaligned action is an interesting direction for future work.
>
> ### 2. Trajectory-Level Awareness (W2 & Q2)
>
> Thanks for this insightful point regarding cumulative, long-term drift. While in our setting the detection decision targets the current action step rather than post-hoc analysis over entire trajectory, the evaluation is inherently context-dependent: understanding an action and predicting its consequences requires the interaction history, as reflected in §2.1 where alignment is defined over $(I, \tau_{<t}, o_t, a_t)$.
>
> The narrative summarizer serves exactly this purpose. By maintaining a running summary of all prior actions and their outcomes, it provides the context for subsequent detection. This implicitly offers defense against cumulative drift, as the evolving trajectory can reveal when accumulated actions diverge from the original intent.
>
> Here is a concrete example from RedTeamCUA trajectories: In a task where an injection directs the agent to repeatedly open terminals to degrade system availability, individual terminal-opening actions appear harmless and useful for subsequent operations, and thus the first 3 such actions pass the guardrail. However, upon the 4th attempt, DeAction recognizes this repetitive history from narrative summary, identifies that "Opening more terminals does not advance the original goal", and flags it.
>
> That said, explicitly designing defense mechanisms against such cumulative drift is a valuable direction for future work.
>
> ### 3. Fine-Tuning Smaller Models (Q1)
>
> This is a highly practical suggestion. As the first effort in misaligned action, no training data existed prior to this work and we design DeAction as a model-agnostic framework that maximizes its plug-and-play applicability. Fine-tuning a smaller, specialized model is a natural next step. MisActBench could serve as a labeled dataset, and the structured outputs from our systematic analysis could serve as high-quality supervision for such distillation. We will include this discussion of future direction in the revision.
>
> ### 4. Limitation Discussion
> Thanks for this suggestion. While our benchmark and guardrail are intended for defensive use, they may help reveal realistic failure modes. However, we believe the benefits substantially outweigh such risks and making detection methods and benchmarks publicly available enables the broader community to collectively improve CUA safety. We will include this discussion in the revised version.

---

> > ### Author Rebuttal · Reviewer_LAzs · 2026-03-31
> >
> > I thank the authors for their clear rebuttal. I appreciate the explanation of how the narrative summarizer implicitly mitigates cumulative drift, and the commitment to discussing fine-tuning smaller models and dual-use risks in the revision.
> >
> > The rebuttal fully resolves my questions. Having considered the score adjustment, I find that my original score remains the most accurate reflection of the paper's solid baseline contribution. Therefore, I am maintaining my positive score of 5.

---

### Official Review · Reviewer_WkuC · 2026-02-23

**Soundness:** 3
**Presentation:** 4
**Significance:** 3
**Originality:** 2
**Overall Recommendation:** 4
**Confidence:** 4

**Summary:**

This paper studies action-level misalignment detection in computer-use agents (CUAs), introduces the MISACTBENCH benchmark, and proposes DEACTION, a two-stage runtime guardrail for detecting and correcting misaligned actions. Experimental results demonstrate that DEACTION achieves strong performance on MISACTBENCH.

**Compliance With Llm Reviewing Policy:**

Affirmed.

**Final Justification:**

The authors' rebuttal has fully addressed all of my concerns.

**Key Questions For Authors:**

1. **On the categorization of misaligned actions.** The paper divides misaligned actions into Malicious Instruction Following, Harmful Unintended Behavior, and Other Task-Irrelevant Behavior. What is the justification for this taxonomy? Are these categories mutually exclusive and collectively exhaustive? In practice, could there be overlapping or hybrid cases that do not cleanly fall into one category?

2. **On benchmark comparison clarity.** Table 1 would be more informative if it also included the scale (e.g., number of trajectories and action steps) of the compared benchmarks, enabling clearer positioning of MISACTBENCH in terms of size and coverage.

3. **On benchmark difficulty and generalization.** Since DEACTION already achieves relatively strong performance on MISACTBENCH, it would be helpful to further analyze the benchmark's difficulty level. For example:

   - How do different state-of-the-art CUA models (without action-level misalignment detection) perform on MISACTBENCH?

   - Does the benchmark remain challenging under stronger backbones or alternative detection strategies?

**Limitations:**

yes

**Strengths And Weaknesses:**

**Strengths**

1. **Well-structured and comprehensive framework.** The paper presents a complete pipeline including problem formulation, benchmark construction, guardrail design, and both offline and online evaluations. The structure is clear and easy to follow.
2. **Practically relevant problem setting.** The focus on action-level misalignment detection is meaningful for real-world CUA deployment.

**Weaknesses**

1. **Limited ecological realism of the benchmark.** While MISACTBENCH provides action-level annotations, internally arising misaligned behaviors are largely constructed through template-based subgoal primitive injection. Although this controlled synthesis strategy enables scalable data generation, it may not fully capture organically emerging, long-tail failure patterns in real-world CUAs.
2. **Limited methodological novelty of DEACTION.** The proposed framework mainly relies on structured prompting and staged LLM reasoning (fast check + systematic analysis + iterative feedback). While the engineering design is careful and practical, the method does not introduce new modeling mechanisms, learning strategies, or theoretical insights. The contribution appears primarily engineering in nature.

---

> ### Author Rebuttal · Authors · 2026-03-31
>
> We thank the reviewer for recognizing the “well-structured and comprehensive framework” and the practicality and meaningfulness of our problem setting.
>
> ### 1. Realism of MisActBench (W1)
>
> We apologize for the unclear presentation that may have led to this inaccurate impression and would like to clarify here:
>
> 36% of internally arising misaligned actions are collected from existing benchmarks without any subgoal primitive injection, while we empirically found that Harmful Unintended Behavior (HUB) occur rarely (only ~⅙). We thus additionally use controlled synthesis as a complement for HUB.
>
> For the trajectories with subgoal injection, all the actions are still generated from actual CUA execution without any human alteration. And all the trajectories further undergo rigorous human screening and validation to ensure plausibility and realism (§A.2).
>
> Please see our response to Reviewer E5dc’s W2 for more. We will revise the presentation in §3.1 to make the things above clearer.
> ### 2. Contribution and Novelty of DeAction (W2)
> We appreciate the recognition of the design and practicability of DeAction, and would like to clarify our contribution more precisely:
>
> Our intent-centric perspective for runtime detection of action-level misalignment in CUAs, a significant shift from safety/policy-centric or retrospective methods in prior work, poses unique challenges: the guardrail must 1) detect under tight *latency constraints* without hindering normal agent execution; 2) reason *pre-execution* without access to actual consequences; 3) analyze *action alignment* rather than defined safety policies.
>
> DeAction addresses these through principled design choices with careful ablations: two-stage detection balancing latency and performance (Tab 4), narrative summaries retaining long-horizon context (Fig 4), and structured alignment analysis beyond generic LLM reasoning (response to Q3).
>
> Beyond the guardrail itself, our work presents the first systematic study of misaligned action detection in CUAs, offering a complete package: the formulation and categorization, the MisActBench benchmark with human-annotated action alignment labels, and DeAction as a strong practical guardrail. Specialized guardrail training is a promising next step, and we believe our work provides a foundation for such efforts in future work.
> ### 3. Categorization Justification (Q1)
> Thanks for bringing this up. Our categorization is operationally exhaustive and exclusive. Any misaligned action falls into exactly one category based on its origin (external/internal) and consequence (harmful/non-harmful).
>
> In practice, among 1,000 annotated misaligned actions, no instance fell outside three categories or exhibited genuine category overlap. 90% received fully consistent labels from all three annotators, and the remaining discrepancies, upon examination, turned out to be divergent interpretations of harm severity rather than overlap, and were resolved by majority voting.
>
> Please see our response to Reviewer E5dc’s W1 for more discussion.
> ### 4. Benchmark Scale Comparison (Q2)
> Thanks for the valuable suggestion. We will include the following scale comparison in the revision.
> |Benchmark|# Instances|
> |-|-|
> |Pre-Exec Bench|1,672|
> |PSG-Agent Benchmark| 2,900|
> |R-Judge|569|
> |ASSEBench|2,293|
> |PolicyGuardBench|59,997 (synthesized from 733 trajectories)|
> |ShieldAgent-Bench|3,110|
> |MobileRisk|204|
> |**MisActBench**|**2,264 (558 trajectories)**|
>
> While some benchmarks include more instances, MisActBench uniquely provides action-level annotations on CUA trajectories, significantly increasing annotation complexity compared to trajectory-level and text-only benchmarks.
> ### 5. Benchmark Difficulty and Generalization (Q3)
> > How do different SOTA CUA models (without action-level misalignment detection) perform on MISACTBENCH?
>
> We'd first like to clarify that MisActBench is designed to evaluate the misaligned action detection ability of models and guardrails, so "without action-level misalignment detection" is not an applicable setting. That said, we are happy to provide results of direct prompting and DeAction with frontier models (including GPT-5.4 and Claude Sonnet 4.5, which demonstrate strong performance on CUA benchmarks).
> |Method|Backbone|Precision|Recall|F1|
> |-|-|-|-|-|
> |Direct Prompting|Claude Sonnet 4.5|61.4|69.9|65.4|
> |DeAction|Claude Sonnet 4.5|83.6|74.8|78.9|
> |Direct Prompting|GPT-5.1|70.4|67.8|69.1|
> |DeAction|GPT-5.1|91.0|71.2|79.9|
> |Direct Prompting|GPT-5.4|72.7|87.9|78.7|
> |DeAction|GPT-5.4|92.0|79.4|85.2|
> > Does the benchmark remain challenging under stronger backbones or alternative detection strategies?
>
> As shown above, our benchmark remains challenging. With the strongest backbone (GPT-5.4), direct prompting can only achieve 78.7% F1, 6.5% lower than DeAction, confirming that MisActBench is not trivially solvable by strong LLMs alone. Even with DeAction, the highest F1 score only reaches 85%, with roughly 20% of misaligned actions still going undetected.

---

> > ### Author Rebuttal · Reviewer_WkuC · 2026-04-01
> >
> > Thank you for the detailed and constructive responses. The additional clarifications on the taxonomy, benchmark scale comparison, and the added experimental results with stronger backbones help address my concerns. In particular, the explanation of the categorization criteria and the empirical annotation consistency provide sufficient justification, and the benchmark scale comparison improves the clarity of Table 1. The additional evaluation with frontier models also helps better position the difficulty of MisActBench.
> >
> > Overall, my questions have been fully addressed. I have updated my score accordingly.

---

### Official Review · Reviewer_E5dc · 2026-03-12

**Soundness:** 3
**Presentation:** 3
**Significance:** 3
**Originality:** 3
**Overall Recommendation:** 5
**Confidence:** 3

**Summary:**

This paper studies misaligned actions in computer-use agents (CUAs) from an intent-centric perspective, arguing that problematic behavior is not limited to explicit policy violations but also includes harmful unintended behavior and task-irrelevant actions. Overall, the paper's fundamental theme concerns whether an agent’s next action can be justified as advancing the user’s authentic goal before execution. The authors seek to examine a significant issue for real-world deployment by introducing MISACTBENCH, a benchmark with 558 trajectories and 2,264 human-annotated action labels across three categories of misalignment, and by proposing DEACTION, a plug-and-play runtime guardrail with a two-stage detector and iterative correction loop. The benchmark combines attack-induced trajectories with synthesized unintended subgoals, and the annotation procedure reports strong agreement.
Empirically, DEACTION improves offline F1 substantially over adapted action-alignment baselines and reduces adversarial attack success rates to 8% across the evaluated CUAs while largely preserving benign task success, with moderate latency overhead.

**Compliance With Llm Reviewing Policy:**

Affirmed.

**Final Justification:**

The authors' rebuttal addresses all my concerns.

**Key Questions For Authors:**

Please see Weaknesses above.

**Limitations:**

There is no discussion of limitation.

**Strengths And Weaknesses:**

**Strengths**

1. The problem formulation is important and well motivated. The paper makes a convincing case that a safety-only framing is too narrow for CUAs, and the shift to action alignment / user-intent preservation is a meaningful conceptual contribution. I found this framing significant because it captures failures that are neither classic attacks nor explicit policy violations, yet still matter a great deal for reliability and trust. This is also one of the paper’s main originality points.

2. MISACTBENCH is a useful benchmark contribution. The benchmark provides action-level labels in multimodal computer-use settings, covers both external attacks and internally arising failures, and includes a reasonably careful annotation pipeline with three annotators and high agreement. Even if the benchmark is not perfect, it fills a clear gap relative to prior trajectory-level safety/policy datasets and should be useful to future work. From a significance standpoint, this dataset contribution alone makes the paper valuable.

3. DEACTION is practical and empirically strong. The two-stage design, narrative summarization, and correction-via-feedback loop form a coherent system rather than a single heuristic. The empirical results are also fairly compelling: offline F1 improves clearly over prior adapted baselines, and online evaluation shows strong adversarial robustness gains with manageable overhead while preserving benign utility. This combination of method design and end-to-end evaluation makes the work sound and relevant to deployment.

**Weaknesses**

1. The benchmark/evaluation still appears somewhat closed-set, despite the paper’s broader motivation. Although MISACTBENCH has action-level annotations for three categories of misalignment, it is unclear what happens when a target agent exhibits behavior that is clearly misaligned but is also outside the current annotation schema, or simultaneously mixes multiple types of misalignment. The rebuttal should clarify whether the taxonomy is meant to be exhaustive, how overlapping cases were handled during annotation, and how the method would behave under open-world, previously unseen forms of misalignment.

2. The internally arising misalignment examples are partly synthesized rather than naturally collected from real agent failures. As a result, these examples may be cleaner and more artificial than how agents actually go off-task in practice. The authors should provide stronger evidence that the constructed failures are realistic and representative of real-world misalignment.

3. The empirical evaluation is good, but some generalization questions remain. Offline comparison is limited to a small number of adapted action-alignment baselines, and the paper acknowledges that internally arising categories remain harder to detect and classify. I would like more discussion of false-positive cost in long benign sessions, robustness to annotation ambiguity, and performance on more organically occurring non-attack failures rather than predominantly benchmarked or synthesized ones. These points do not invalidate the results, but they matter for judging how far the current evidence supports the claim of a “universal” runtime guardrail.

4. Related-work positioning should better discuss adjacent intent-centric work. For example, [1][2] also centers the problem of user-intent preservation and erroneous planning, though in API-calling LLM agents rather than computer-use agents.

[1] Feng et al. TAI3: Testing Agent Integrity in Interpreting User Intent. NeurIPS'25. https://arxiv.org/abs/2506.07524

[2] Ji et al. Testing and Understanding Erroneous Planning in LLM Agents through Synthesized User Inputs. https://arxiv.org/abs/2404.17833

---

> ### Author Rebuttal · Authors · 2026-03-31
>
> ### 1. Category Exhaustiveness and Overlapping Cases (W1)
>
> > whether the taxonomy is meant to be exhaustive; if not, how the model would behave under unseen forms of misalignment.
>
> Thanks for bringing this up. To clarify, we do target comprehensive benchmarking and evaluation. We would love to take this chance to further explain why our categorization is exhaustive rather than close-set.
>
> Any misaligned action falls into exactly one category based on its origin (external/internal) and consequence (harmful/non-harmful): If the action is misaligned due to external redirection, it falls into Malicious Instruction Following; otherwise, it falls either into Harmful Unintended Behavior (exposing safety risks) or Other Task-Irrelevant Behavior (degrading reliability and efficiency) based on its consequence severity.
>
> > overlapping cases during annotation
>
> As described above, the categories are non-overlapping by construction. In practice, among 1,000 annotated misaligned actions, no instance fell outside three categories or exhibited genuine category overlap. 90% received fully consistent labels from all three annotators, and the remaining discrepancies, upon examination, turned out to be divergent interpretations of harm severity rather than overlap, and were resolved by majority voting.
>
> In conclusion, our categorization is designed to be exhaustive and does cover all the possible real-world scenarios.
> ### 2. Realism of Benchmark Data (W2)
>
> Thanks for raising this critical concern around balancing scalability with realism.
>
> First, we want to apologize for the unclear presentation in §3.1 that may have led to an inaccurate impression, and would like to clarify here:
>
> A substantial part (36%) of internally arising misaligned actions in MisActBench are actually collected from existing benchmarks without any subgoal primitive injection. During the collection, we found that empirically Harmful Unintended Behavior (HUB) actions occur rarely among these uncontrolled trajectories (only ~⅙  out of those actions). To ensure sufficient coverage for comprehensive benchmarking, we additionally use controlled synthesis as a complement for HUB.
>
> We would also like to note that, for the trajectories with subgoal injection, all the actions are still generated from actual CUA execution without any human alteration. And all the trajectories further undergo rigorous human screening and validation to ensure plausibility and realism (§A.2).
>
> In summary, controlled synthesis serves mainly to ensure sufficient coverage of the underrepresented HUB category, with all synthesized trajectories grounded in real agent execution and validated by human reviewers. We will revise the presentation in §3.1 to make this clearer.
>
>
> ### 3. Generalization and False-Positive Cost (W3)
> Thanks for the suggestion. We analyzed 20 randomly sampled trajectories from OSWorld evaluation (252 steps). DeAction flagged 47 steps, including 14 false positives. Upon closer examination, all of them were effectively harmless: the agent either insisted or proposed an aligned alternative via the correction loop, or performed harmless fallbacks (e.g., waiting, clicking empty areas). This echoes Tab 5, where DeAction preserves or slightly improves success rates across CUAs in benign sessions.
>
> We’d also like to note that our OSWorld online evaluation (§5.2) directly evaluates DeAction on organically occurring misaligned actions during live execution in benign, non-attack setting, serving as a complementary validation to our offline evaluation.
>
> Regarding annotation ambiguity, our human annotation achieved a Fleiss' Kappa of 0.84 (near-perfect agreement) on MisActbench.
>
> ### 4. Related Work (W4)
> Thanks for pointing out these related works on API-calling LLM agents. We will discuss them in the revision. Beyond the domain differences, while they similarly adopt an intent-centric perspective, they focus on stress-testing agents with synthesized inputs and post-hoc analysis. In contrast, we focus on *runtime*, *pre-execution* detection and correction of misaligned actions during *live* execution.
>
>
> ### 5. Limitation
> Thanks for pointing this out. We will add a limitation section discussing our error analysis (§E.2), synthesis pipeline constraints and generalization challenges to future, unknown misalignments.

---

> > ### Author Rebuttal · Reviewer_E5dc · 2026-04-05
> >
> > Thanks author for rebuttal. I raise score to 5. Please do revision as suggested.

---

### Official Review · Reviewer_V9uF · 2026-03-13

**Soundness:** 3
**Presentation:** 3
**Significance:** 3
**Originality:** 2
**Overall Recommendation:** 5
**Confidence:** 2

**Summary:**

This paper addresses the problem of misaligned actions in computer-use agents (CUAs), actions that deviate from the user's authentic intent during automated GUI interaction. The authors propose an intent-centric framework that goes beyond traditional safety-focused approaches by identifying three categories of misalignment. They introduce MisActBench, a benchmark of 558 trajectories with 2,264 human-annotated action-level alignment labels, constructed via a hybrid pipeline combining adversarial benchmark execution and controlled trajectory synthesis. They further propose DeAction, a two-stage runtime guardrail that first performs a lightweight fast check to pass clearly aligned actions, then applies systematic analysis (injection analysis, action understanding, outcome prediction, misalignment analysis) for ambiguous cases. DeAction also provides structured corrective feedback to iteratively guide the agent toward aligned behavior. Experiments show DeAction outperforms baselines by over 12% F1 on MisActBench offline evaluation, and reduces attack success rates by over 75% in online settings while preserving or improving benign task success rates.

**Compliance With Llm Reviewing Policy:**

Affirmed.

**Final Justification:**

I am working with the program chair and field chair to resolve my error and ensure your paper receives a fair evaluation.

**Key Questions For Authors:**

Several aspects of the experimental setup lack sufficient detail:

(a) How are the narrative summaries evaluated for quality? Poor summaries could propagate errors.

(b) What is the distribution of misalignment categories in the online evaluation benchmarks?

(c) The paper mentions using "lighter counterpart" models for fast check but does not systematically evaluate the sensitivity to this choice.

(d) The maximum correction iterations are set to 3 without justification, so what happens with more or fewer iterations?

**Limitations:**

There is a limitations discussion in the form of failure case analysis (Section 5.1, error analysis) and mention limitations of the fast check being deceivable, grounding failures in coordinate-based actions, and flawed intent-level reasoning.

**Strengths And Weaknesses:**

Strengths:

1. The shift from a purely safety/policy-violation lens to an intent-centric action alignment perspective is a meaningful conceptual contribution.

2. MisActBench is carefully constructed through a hybrid pipeline that combines real adversarial trajectories from four existing benchmarks with synthesized trajectories for internally arising misalignment.

3. The evaluation is thorough, spanning both offline detection on MisActBench and online end-to-end evaluation on RedTeamCUA and OSWorld.

4. The two-stage detection pipeline (fast check + systematic analysis) is a pragmatic design that balances accuracy and latency.

Weaknesses:

1. While the problem framing and benchmark are novel, the DeAction method itself is a relatively straightforward application of LLM-based reasoning. The fast check is essentially a binary classification prompt, the systematic analysis is a structured chain-of-thought prompt with four predefined analysis dimensions, and the iterative correction feeds back the analysis as a corrective prompt. The contribution lies more in the specific prompt engineering and the combination of components.

2. A significant portion of MisActBench (22% of trajectories per Figure A.2) comes from synthesized trajectories where misaligned actions are artificially injected into benign trajectories. While the authors describe manual validation, the synthesis procedure raises concerns: (a) the injected misaligned actions may not fully capture the subtlety and diversity of real-world internally arising misalignment, since they are generated from a fixed set of 13 unintended subgoal primitives (Table A.1); (b) the merging process at a single injection point creates a somewhat artificial pattern where misalignment occurs in a concentrated burst rather than sporadically as in real deployment. The paper would benefit from analysis of how detection performance differs between real adversarial trajectories and synthesized ones.

---

> ### Author Rebuttal · Authors · 2026-03-31
>
> Dear Reviewer V9uF,
>
> Thank you for your time and effort in reviewing.
>
> However, we noticed that the content of your review appears to correspond to a different submission.
>
> Specifically, your review refers to a paper about off-task actions in reinforcement learning with MDP formulation, RL algorithm designs, grid-world/Crafter experiments, etc. In contrast, our submission (#17652) focuses on misaligned action detection in computer-use agents, including an evaluation benchmark MisActBench and a runtime guardrail DeAction.
>
> We kindly request that you check if the review was uploaded to the correct submission ID. We would greatly appreciate it if you could review our actual submission and provide your feedback.
>
>
> Best,
>
> Authors of Submission 17652

---

> > ### Author Rebuttal · Reviewer_V9uF · 2026-04-02
> >
> > Dear Authors of Submission 17652,
> >
> > I sincerely apologize for the mix-up. I was reviewing two submissions that both involve off-task actions, and mistakenly posted the review for the other paper to your submission. I am already working with the Program Chairs and Area Chair to resolve this, and will make sure your paper receives a fair evaluation.
> >
> > Sorry again for the inconvenience.
> >
> > Best regards,
> > Reviewer V9uF

---

> > > ### Author Response · Authors · 2026-04-04
> > >
> > > We sincerely thank the reviewer for the positive assessment and for recognizing that our **intent-centric perspective is "a meaningful conceptual contribution"**, that **MisActBench is "carefully constructed"**, that our **offline and online evaluation is "thorough"**, and that the DeAction is **"a pragmatic design that balances accuracy and latency"**.
> > >
> > > ### 1. Contribution of DeAction (W1)
> > >
> > > While DeAction leverages LLM-based reasoning, we would like to clarify our contribution more precisely:
> > >
> > > Our shift poses unique challenges: *latency constraints*, *pre-execution* reasoning and *action alignment* focus. DeAction addresses these through principled design choices with careful ablations: two-stage detection balancing latency and performance (Tab 4), narrative summaries retaining long-horizon context (Fig 4), and structured alignment analysis beyond generic LLM reasoning (see response to WkuC’s Q3).
> > >
> > > Beyond the guardrail, our work presents the first systematic study of misaligned action detection in CUAs, offering a complete package and providing a foundation for future work such as specialized guardrail training.
> > >
> > > Please see our response to Reviewer WkuC’s W2 for more.
> > >
> > > ### 2. Concern on Synthesis Data (W2)
> > >
> > > We appreciate this thoughtful concern. We want to first apologize for the unclear presentation in §3.1 and clarify here: 36% of internally arising misaligned actions are collected from existing benchmarks without any subgoal injection, while we empirically found that Harmful Unintended Behavior (HUB) occur rarely (only ~⅙). We thus additionally use controlled synthesis as a *complement* for HUB, with actual CUA execution and human validation for realism (§A.2). Please see our response to Reviewer E5dc's W2 for more.
> > >
> > > Following the suggestion, we also report DeAction's recall on internally arising misalignment categories, separated by source:
> > >
> > > |Split| Recall |
> > > |-|-|
> > > |Collected trajectories|66.5|
> > > |Synthesized trajectories|72.5|
> > >
> > > The gap between the two sources is moderate, and misaligned actions on synthesized trajectories are not trivially detectable. This confirms that controlled synthesis produces challenging evaluation data rather than artificially easy cases, serving its intended role as a complement for comprehensive benchmarking.
> > >
> > > ### 3. Quality of Narrative Summary (Q1)
> > >
> > > While there is a possibility of summarization errors propagating, this might not be a practical concern:
> > >
> > > * The summarization task here is relatively simple: it describes a single $(o_t, a_t, o_{t+1})$ transition rather than requiring complex multi-step reasoning. We conduct an ablation comparing GPT-5.1 and GPT-5 mini as the summarizer, and performance is nearly identical, confirming that this task is straightforward enough to be performed reliably:
> > > |Summarizer|Precision|Recall|Acc|F1|
> > > |-|-|-|-|-|
> > > |GPT-5.1|86.5|74.3|84.6|79.9|
> > > |GPT-5 mini|88.2|73.8|85.2|80.4|
> > >
> > > * Our ablation in Fig 4 provides further validation. Narrative summaries outperform raw screenshots as history representation, indicating that summaries retain sufficient long-horizon context for alignment assessment while filtering out visual noise.
> > >
> > > ### 4. Distribution in Online Evaluation Benchmark (Q2)
> > >
> > > Thanks for this suggestion. As online evaluation is based on live interactive execution, we do not have exhaustive step-level annotations for every rollout. To provide a rough distribution, we randomly sample 15 trajectories with flagged actions from each setting and manually label them:
> > >
> > > |Setting|MIF|HUB|OTIB|
> > > |-|-|-|-|
> > > |RedTeamCUA |76.5%|0%|23.5%|
> > > |OSWorld|0%|6.7%|93.3%|
> > >
> > > The distribution aligns with each setting’s design. RedTeamCUA primarily exposes Malicious Instruction Following as expected from its prompt injection setup, while benign setting (OSWorld) only exhibits internally arising misalignment. Harmful Unintended Behavior is the rarest in both settings, echoing our justification of using controlled synthesis as a complement to ensure coverage in MisActBench.
> > >
> > > ### 5. Fast Check Model Choice (Q3)
> > > As fast check serves as a conservative screening for clearly aligned actions, its decisions are inherently easier. Ambiguous cases are routed to the systematic analysis stage, where model capability matters most.
> > >
> > > To further resolve the question, we additionally conduct an ablation replacing the fast check model as below, indicating that using GPT-5.1 or a lightweight GPT-5 mini has virtually no impact on detection performance :
> > >
> > > |Fast Check Model|Precision|Recall|Acc|F1|
> > > |-|-|-|-|-|
> > > |GPT-5.1|87.8|74.3|85.2|80.5|
> > > |GPT-5 mini|88.2|73.8|85.2|80.4|
> > >
> > > ### 6. Maximum Correction Iteration (Q4)
> > > The maximum iteration count is a configurable parameter. We set the default to 3, as empirically it corrects most cases (81%) while keeping latency manageable (§5.2, L405-410). In real-world deployment, it can be tuned based on application requirements: alignment-critical settings may allow more iterations, while latency-sensitive ones may reduce it.

---

### Decision · Program_Chairs · 2026-04-30

**Decision:**

Accept (regular)

**Comment:**

This paper presents MISACTBENCH and DEACTION for detecting misaligned CUA actions. All reviewers unanimously lean towards acceptance. Initially, reviewers raised concerns about DEACTION's limited algorithmic novelty and the realism of synthesized trajectories. In the rebuttal, authors clarified that 36% of internal misalignments were organically collected, justifying synthesis for rare harmful behaviors. They also showed the benchmark remains challenging for frontier models (GPT-5.4 achieved 78.7% F1). Concerns over cumulative drift were addressed via the narrative summarizer’s contextual tracking. Given the broad reviewer support, I recommend acceptance of the paper.